# CooHOI: Learning Cooperative Human-Object Interaction with Manipulated Object Dynamics

**Jiawei Gao** [*1,2], **Ziqin Wang** [*1,3], **Zeqi Xiao**[1,4], **Jingbo Wang**[1], **Tai Wang**[1], **Jinkun Cao**[5],
**Xiaolin Hu**[2], **Si Liu**[†3], **Jifeng Dai**[†2], **Jiangmiao Pang**[†1]
[1]Shanghai AI Laboratory, [2]Tsinghua University, [3]Beihang University,
[4]Nanyang Technological University, [5]Carnegie Mellon University,

## Abstract

Enabling humanoid robots to clean rooms has long been a pursued dream within humanoid research communities. However, many tasks require multi-humanoid collaboration, such as carrying large and heavy furniture together. Given the scarcity of motion capture data on multi-humanoid collaboration and the efficiency challenges associated with multi-agent learning, these tasks cannot be straightforwardly addressed using training paradigms designed for single-agent scenarios. In this paper, we introduce **Coo**perative **H**uman-**O**bject **I**nteraction (**CooHOI**), a framework designed to tackle the challenge of multi-humanoid object transportation problem through a two-phase learning paradigm: individual skill learning and subsequent policy transfer. First, a single humanoid character learns to interact with objects through imitation learning from human motion priors. Then, the humanoid learns to collaborate with others by considering the shared dynamics of the manipulated object using centralized training and decentralized execution (CTDE) multi-agent RL algorithms. When one agent interacts with the object, resulting in specific object dynamics changes, the other agents learn to respond appropriately, thereby achieving implicit communication and coordination between teammates. Unlike previous approaches that relied on tracking-based methods for multi-humanoid HOI, CooHOI is inherently efficient, does not depend on motion capture data of multi-humanoid interactions, and can be seamlessly extended to include more participants and a wide range of object types.

## 1 Introduction

Imagine the scenario where you're moving home and seeking help from humanoid robots. However, certain items, like beds and sofas, are too large and heavy for a single robot to manage effectively. Despite significant advancements in learning and control of physics-based human-object interactions [6, 43, 21, 37, 20, 45, 7] and humanoid robots performing complex operations [29, 19, 26, 2, 42, 41, 10, 9, 39], the area of **multiple humanoids collaboratively transporting objects**—particularly when a single individual may find it challenging to handle heavy or long items—remains relatively under-investigated. Some previous efforts [44] tried to address these challenges using tracking-based methods, capturing interactions between two characters during tasks like box carrying, and training policies to mimic these actions. However, obtaining comprehensive motion data on interactions involving multiple participants is costly, and tracking-based methods struggle to adapt to scenarios with varying object sizes and an increased number of agents. Another straightforward idea might be to directly train a multi-agent policy from scratch, using an approach

---

[*]Equal contributions. Email: winstongu20@gmail.com, wzqin@buaa.edu.cn
[†]Corresponding author. Email: pangjiangmiao@gmail.com

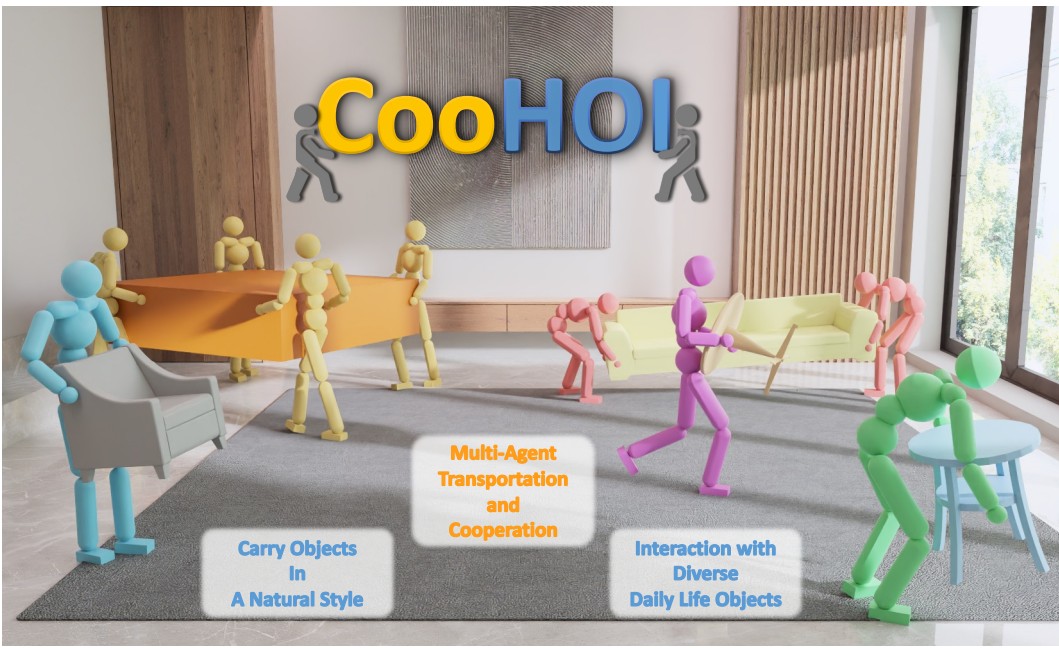

Figure 1: Our framework empowers physically simulated characters to execute multi-agent human-object interaction (HOI) tasks with naturalness and precision.

similar to that for single agents. However, the expanded action sampling space significantly slows down the overall training process, preventing the policy from converging properly.

Inspired by how humans learn cooperation skills, *i.e.* beginning with mastering tasks independently and then developing strategies for collaborative efforts in coordination tasks, we propose a two-stage framework for training multi-agent cooperation strategies, named **Coo**perative **H**uman-**O**bject **I**nteraction, **CooHOI**. In the initial stage, we train a single-agent object carrying policy utilizing the AMP framework [25, 7]. To ensure agents focus significantly on the **dynamics** of the objects they are tasked with transporting, we include the state and velocity information of the object's bounding box in the agent's observation space. The second stage aims to transfer these individual carrying skills into collaborative strategies. When two humans collaborate to carry a long object, they typically hold it at opposite ends. Moreover, when one individual takes action, it affects the **dynamics** of the object, enabling the other individual to perceive this change and adjust their actions accordingly for effective cooperation. Therefore, we adjust the observation for both agents to focus on the bounding boxes located at the ends of the long objects. This refinement aligns with the observation space used during the single-agent training phase, effectively leveraging the previously developed single carrying skills. Additionally, the inherent rigid body characteristic of long objects facilitates implicit communication between agents. This setup allows an agent to adeptly adjust to their teammate's actions by observing changes in the object's **dynamics**, thereby enhancing coordination and cooperation in carrying tasks. The overall framework is illustrated in Figure 2.

To validate the effectiveness of our framework, we conducted experiments where we trained control policies for two humanoid characters to carry various long objects, such as boxes and sofas. Our results demonstrate that our framework enables these characters to exhibit natural-looking behaviors while successfully completing cooperative tasks, utilizing only motion capture data from one single agent. We compared our approach against the baseline method of training from scratch and performed detailed ablation studies to evaluate the impact of our design decisions, also testing the limitations of our framework. In summary, our main contributions are as follows: 1)We have developed an efficient and robust framework, **CooHOI**, for training physically simulated characters in cooperative object transporting tasks, demonstrating significant effectiveness. 2)We have established that utilizing object dynamics for communication proves to be an effective strategy in learning cooperative object transporting tasks.

## 2 Related Work

**Physics-based Human-Object Interaction Motion Synthesis.** Synthesizing natural and physically plausible human-scene interactions, such as humanoid characters sitting on chairs, lying on beds, and carrying boxes, is crucial for advancements in character animation and robotics. Physics-based methods leverage physics simulators [18, 35] to control characters modeled as interconnected rigid bodies through joint torques and deep reinforcement learning methods. To facilitate the training process, some strategies employ tracking-based techniques [16, 23, 1, 38, 20, 44], which rely on the availability of high-quality reference motions. This reliance often limits their application in human-scene interactions due to the scarcity of suitable data and affects their versatility across different scenarios. Recently, the Adversarial Motion Priors(AMP) framework [25] introduced the use of a discriminator to ensure that generated motions align with the distribution of reference motions. This approach has shown success in various downstream tasks [11, 24, 34], including human-scene interactions [7, 21, 37]. Despite these advancements, there has been limited focus on synthesizing cooperative behaviors among multiple characters interacting with objects—a gap that our work aims to address.

**Multi-Character Control.** While significant advancements have been made in synthesizing motions for single agents, the realm of multi-character animation remains relatively unexplored. Existing approaches predominantly rely on kinematic-based methods [36, 14, 32, 30, 5] and datasets of multi-character interactions [13, 31, 4]. These methods, however, require high-quality interaction data and often fall short in ensuring physical plausibility or adequately handling multi-character cooperative interactions with objects. Physics-based techniques for character animation have primarily focused on aspects like crowd navigation [8, 27], limiting themselves to behaviors such as pedestrian movement and collision avoidance. Although [44] showcases interactions among multiple characters and object manipulation, such as carrying long boxes, these tracking-based approaches necessitate motion capture data of human interactions or human-object interactions and struggle to scale to an arbitrary number of agents or different types of objects. In contrast, our framework requires only single agent motion capture data for multi-character object transporting tasks and can easily extend to different types of objects and different numbers of agents.

## 3 Methodology

Figure 2 shows the overall framework of our approach. Our method utilizes motion data from individual agents and involves a two-stage learning process to create cooperative control policies. First, we train a policy for single-agent carrying tasks, using the dynamics of the manipulated object

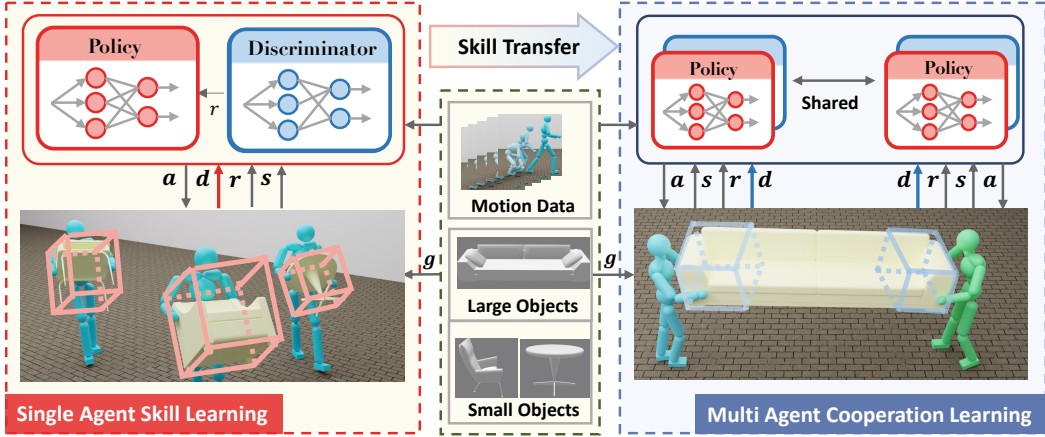

Figure 2: Our framework employs a two-phase learning paradigm. In the first phase, depicted on the left, we train single-agent carrying skills by imitating from human motion priors. In the second phase, we transfer these single-agent skills to a cooperative context. Notably, we use the dynamics of the object as feedback information, as illustrated by the bounding box shown in the figures.

as feedback, as described in Section 3.2. Then, we use parallel training to develop cooperative strategies, with changes in the object's dynamics acting as a form of implicit communication. This is further explained in Section 3.3.

## 3.1 Preliminary

**Physics-based Character Control.** We formulate physics-based character control as a goal-conditioned reinforcement learning task. At each time step $t$, the agent samples an action from its policy $\pi(a_t \mid \mathbf{s}_t, \mathbf{g}_t)$ based on the current state $\mathbf{s}_t$ and the task-specific goal feature $\mathbf{g}_t$. When this action is applied to the character, the environment transitions to the next state $\mathbf{s}_{t+1}$, and the agent receives a task reward $r^G(\mathbf{s}_t, \mathbf{g}_t, \mathbf{s}_{t+1})$. To train control policies that enable characters to achieve high-level tasks in a natural and life-like manner, we adopt the AMP framework [25]. While this framework aims to optimize the expected cumulative task reward $J(\pi)$, it introduces a discriminator to encourage the character to produce behaviors similar to those in the dataset by providing a style reward $r^S(\mathbf{s}_t, \mathbf{s}_{t+1})$. The agent's reward $r_t$ at each time step $t$ is defined by $r_t = w^G r^G(\mathbf{s}_t, \mathbf{g}_t, \mathbf{s}_{t+1}) + w^S r^S(\mathbf{s}_t, \mathbf{s}_{t+1})$. More details can be found in our appendix and the original AMP paper [25].

**Multi-Agent Reinforcement Learning.** We formulate our cooperative task as a Partially Observable Markov Decision Process (POMDP) [15]. A POMDP with $n$ agents is defined by $\{\mathcal{S}, \mathcal{A}_1, \cdots, \mathcal{A}_n, \mathcal{O}_1, \cdots, \mathcal{O}_n, \mathcal{R}, \mathcal{P}, \mathcal{T}\}$, where $\mathcal{S}$ is the state space, $\mathcal{A}_i$ is the action space for agent $i$, $O_i$ is the local observation for agent $i$, and $\mathcal{R}$ is the shared reward function. Each agent uses its own policy $\pi_\theta(\mathbf{a}_i \mid \mathbf{o}_i)$ to take action $\mathbf{a}_i \in \mathcal{A}_i$ based on its local observation $\mathbf{o}_i \in O_i$. The environment transitions according to the function $\mathcal{P}(\mathbf{s}_{t+1} \mid \mathbf{s}_t, \mathbf{a}_1, \cdots, \mathbf{a}_n)$, where $\mathbf{s}_t, \mathbf{s}_{t+1}$ are states of time step $t$ and $t+1$, respectively. The agents then get a reward $\mathbf{r}_t$ based on the states $\mathbf{s}_t$ and $\mathbf{s}_{t+1}$. The goal of multi-agent reinforcement learning algorithms is to jointly optimize the discounted accumulated reward $J(\theta) = \mathbb{E}_{\mathbf{a}_1^t, \cdots, \mathbf{a}_n^t, s^t}\left[\Sigma_{t=0}^T \gamma^t \mathbf{r}_t\right]$.

## 3.2 Single Agent Carrying Skills Training

In developing our approach for single-agent object manipulation, we introduce several advancements based on previous methods. We integrate the dynamics of the manipulated object into the observation space and introduce a reward function framework for object manipulation tasks. These enhancements allow the trained policy to easily adapt to multi-agent settings.

### 3.2.1 Enriched Goal Feature with Manipulated Object Dynamics as Feedback.

For successful object carrying tasks, we emphasize the critical role of using the dynamics of the manipulated object as feedback. These dynamics are captured through the eight vertices of the object $o$'s bounding box $b_t^{\text{ver}}$, its rotation angle $b_t^{\text{facing}}$, its velocity $b_t^v$ and its angular velocity $b_t^w$, as described in Equation (1).

$$\mathcal{D}_t = \text{concatenate}(b_t^{\text{ver}}, b_t^{\text{facing}}, b_t^v, b_t^w). \tag{1}$$

By incorporating these dynamics information into the observation, we establish a feedback mechanism that keeps agents continually informed about the outcomes of their actions. This also equips agents with the capability to react appropriately, whether engaged in single-agent tasks or collaborative multi-agent environments. Along with the state of the agent $s_t$ and the position of the target $d_t^{\text{pos}}$, we formulate the dynamics-aware task observation as:

$$\mathbf{o}_t = \text{concatenate}(s_t, d_t^{\text{pos}}, \mathcal{D}_t). \tag{2}$$

### 3.2.2 Enriched Task Design facilitating Efficient Skill Transfer.

To facilitate the transition from single-agent to multi-agent object carrying tasks, we decompose the carrying process into three sub-tasks: walking towards the objects, lifting the object from the ground, and carrying the object to its intended destination. Consequently, the reward system is structured into three components: $r_{\text{walk}}$, $r_{\text{held}}$ and $r_{\text{target}}$.

To encourage the agents to choose the face of the object they will face while carrying, we introduce an additional goal feature called **stand points** $x_t^{\text{stand}}$. During training, these stand points are randomly allocated to positions directly in front of the various faces of the object at time step $t$. This strategy is

designed to facilitate multi-agent cooperative training, helping the agents learn to avoid walking to the long side of the object where it is difficult to grasp and carry. We then define $r_{\text{walk}}$ as the distance between the agent and the stand point, as specified in Equation (3).

$$r_{\text{walk}} = \exp\left(\|x_t^{\text{root}} - x_t^{\text{standing}}\|^2\right) \tag{3}$$

Additionally, in scenarios where multiple agents are transporting an object, the object's considerable size often makes it difficult for agents to find an appropriate grip point for lifting. To address this, we introduce a novel concept called **held points**. Specifically, we select the geometric center $h_t$ of each end of the object as the held point and encourage the agents to interact at these points. The reward function $r_{\text{held}}$, as defined in Equation (4), uses $x_t^{\text{hand}}$ to represent the mean position of the agent's two hands. This design aids the transition from individual to collective task environments by encouraging agents to identify the held points at the geometric center of each end of the long object.

$$r_{\text{held}} = \exp\left(\|x_t^{\text{hand}} - h_t\|^2\right) \tag{4}$$

We then define $r_{\text{target}}$ to encourage agents to carry the object to the destination:

$$r_{\text{target}} = \exp\left(\|x_t^{\text{target}} - x_t^{\text{object}}\|^2\right). \tag{5}$$

### 3.3 Cooperation Strategy Training

Upon mastering single-agent tasks involving object carrying, it is crucial to effectively transfer and further enhance these abilities to boost the cooperative learning process. Initially, we facilitate the skill transfer by guiding each agent to lift and transport one end of a large object. Subsequently, we refine the control policy through the application of the Multi-Agent Proximal Policy Optimization (MAPPO) algorithm [40], leveraging the dynamics of manipulated objects for feedback and implicit communication.

#### 3.3.1 Efficient Skill Transfer Using Dynamics Information.

Initially, we replicate the single-agent object-carrying policy for all agents and then fine-tune it within a cooperative framework. However, handling a long object poses a challenge due to its size, which complicates the agents' ability to identify suitable lifting points, thereby hindering the efficient application of their object-carrying skills. To address this issue, we encourage agents to observe the dynamics information—specifically, the state and velocity data of the bounding box at each end of the long object, as outlined in Section 3.2.1. This approach allows agents to simulate carrying a smaller box positioned at the long object's ends, using the dynamics of this "smaller box" as **feedback**, similar to the single-agent skill training process. Additionally, we incorporate previous methodologies, such as stand points and held points described in Section 3.2.2, to enhance the smooth transition of skills.

Moreover, this design of dynamics as observation acts as **implicit communication channel** between agents. When an agent takes an action, the object dynamics will change, and because we use the manipulated object dynamics as feedback in single-agent training, the change in object dynamics will result in a corresponding change in strategy for the other agents. This method of implicit communication presents a straightforward yet effective approach for enhancing teamwork in multi-agent settings. Furthermore, this framework proves to be adaptable with variations in the number of participating agents.

#### 3.3.2 Cooperation Training using CTDE Scheme.

The non-stationary environment of multi-agent reinforcement learning, coupled with sample efficiency issues, presents a significant challenge in training agents to collaborate effectively. During the cooperation training phase, we continue to use a reward function similar to that used in the single-agent training phase and employ the centralized training and decentralized execution (CTDE) [15] scheme for coordination training. During the training phase, we utilize the Multi-Agent Proximal Policy Optimization (MAPPO) algorithm [40] to develop cooperative strategies among agents. This approach involves updating the value function network $V_\phi$ according to Equation (6), using the trajectories $\mathcal{D}$ that are accumulated and shared among all agents. More details can be found in our

Table 1: This table presents our results for single-agent and two-agent carrying of the Box object. "CooHOI" refers to the policy trained using the complete CooHOI framework in both single-agent and two-agent settings. "CooHOI+WeightAug" indicates that we applied the same weight augmentation design as InterPhys [7].

| Agent Number | Methods | Weight (kg) | Distance(m) | Success Rate(%) | Precision (cm) |
|---|---|---|---|---|---|
| Single Agent | InterPhys [7] | [5,26] | [1,10] | 94.3 | 8.3 |
| | CooHOI+WeightAug(Ours) | [2,26] | [1,20] | 93.98 | **4.8** |
| | CooHOI(Ours) | [2,13] | [1,20] | **96.48** | 6.9 |
| Two Agent | From Scratch | [15,40] | [2,20] | 0 | NAN |
| | CooHOI(Ours) | [15,40] | [2,20] | **89.54** | **3.86** |

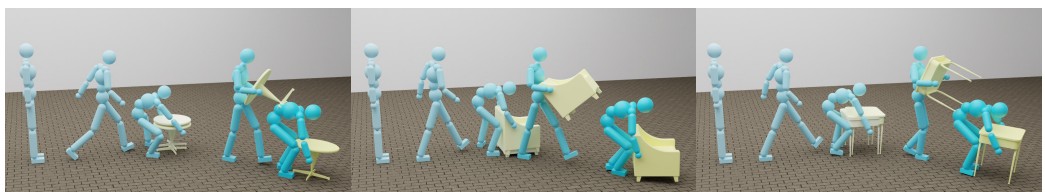

Figure 3: Carrying performance for objects of different categories. From left to right: Table, Armchair, and High Stools. All objects were required to be moved to a location 4 meters away.

appendix and the original MAPPO paper [40].

$$\phi_{k+1} = \arg\min_{\phi} \frac{1}{|\mathcal{D}_k| T} \sum_{\tau \in \mathcal{D}_k} \sum_{t=0}^{T} \left( V_\phi \left( o_t, s_t, \mathbf{a_t}^- \right) - \hat{R}_t \right)^2 \tag{6}$$

In addition, given the homogeneous roles of agents in the collaborative task of carrying long objects, we adopt the strategy of parameter sharing, a method proven to enhance performance across various cooperative tasks [3, 33, 40]. Specifically, we share the parameters of both the policy and value networks among all agents to improve the training of cooperative behaviors.

## 4   Experiments

We conducted extensive experiments to test the effectiveness and also the boundary of capabilities of our framework. The basic experiment setups are explained in Section 4.1 and in Appendix. We evaluate our framework on various object-carrying tasks in Section 4.2. To better understand the importance of different design decisions in our framework, we performed extensive ablation studies in Section 4.3. Since our method primarily focuses on interactions between characters and objects, we also provide extensive visual analysis and presentations to demonstrate our framework.

### 4.1   Experiments Setup

**Datasets and Initialization.**   Our primary source of motion data is the AMASS dataset [17], which provides motions encoded in SMPL [22] parameters. We collected a total of four types of basic reference motion data, including 9 motions related to walking, 5 related to picking up, 4 related to carrying, and 5 related to putting down. To enhance the robustness of the carrying process, we randomly initialized the weight and size of the object, as well as the distance from the person to the destination. Specifically, for a single individual, the object's weight ranged from 5KG to 25KG, its size varied between 0.5 to 1.5 times its original scale, and the distances between the agent and the object, as well as between the object and the destination, ranged from 1 m to 20 m. To enhance the robustness of the carrying process, we randomly varied the weight, size, and of the object, as well as the distance from the person to the endpoint. Furthermore, considering that the process of multiple people carrying might involve situations where someone walks backward, we ensured that the single agent mastered the basic movements for multi-person collaboration by introducing additional motion data involving backward and side walking. More training details can be found in the appendix.

Table 2: The trained policy exhibits the ability to handle various object categories encountered in daily life with simple fine-tuning. We tested the performance of our policy model across different objects.

| Agent Number | Object Category | Distance(m) | Success Rate(%) | Precision (cm) |
|---|---|---|---|---|
| Single Agent | Table | [1,20] | 97.07 | 5.23 |
| | Armchair | [1,20] | 97.26 | 5.05 |
| | HighStool | [1,20] | 99.21 | 4.02 |
| Two Agents | Sofa | [2,20] | 84.17 | 10.12 |

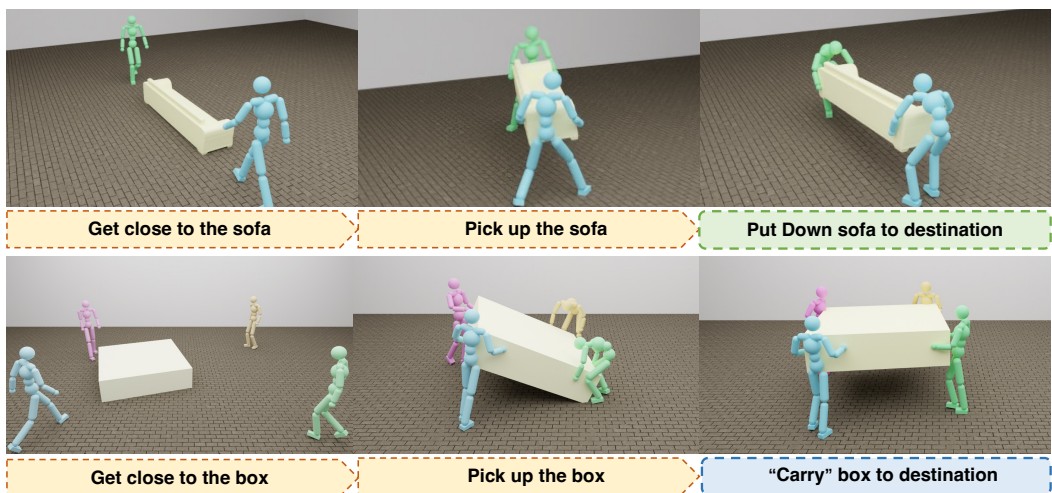

Figure 4: Visualization of cooperative carrying in the multi-agent scenario.

**Metrics.** Following InterPhys [7], we utilize **Success Rate** and **Precision** as the primary metrics. Specifically, a task is considered successful if the distance between the object and its destination is less than 0.2m. Precision measures the distance between the object and the target point across all completed tasks. Moreover, all evaluations are based on average results across 4096 environments and 4 random seeds.

## 4.2 Evaluation on Object Carrying Tasks

**Comprehensive Evaluation on Box Carrying.** We evaluate our policy in both single-agent and two-agent scenarios. Table 1 presents the performance statistics for the carrying task. Consistent with the settings described in Interphys [7], we randomize the weight and distance as outlined in Section4.1. It is important to note that InterPhys [7] includes the weight information of objects as part of the observation, enabling it to achieve a broader weight range. To ensure result comparability, as shown in Line 2, we also incorporated this information into individual training. The results indicate that, with a similar success rate, we increased the transportation distance and improved transportation accuracy, even when using discrete motion data rather than the costly whole-body motion capture data used in InterPhys [7]. As shown in Line 3, although removing the weight information reduces the weight range the agent can handle, it represents a more realistic setting and provides comparable baseline results for future research. Lines 4-5 show the results in multi-agent scenarios. Without using our CooHOI framework and simply employing parallel training for multi-agent tasks, the training fails. More analysis and visualizations will be provided in the following sections.

**Extend to Diverse Daily Life Objects.** Moreover, we are not satisfied with merely transporting simple boxes. Therefore, we have sampled nearly 40 common everyday objects from [6], including three types of single-person objects—Table, Armchair, and HighStool—as well as one type of two-person object, the Sofa, as illustrated in Figure 3 and Line 1 in Figure 4. Additional visualizations of these objects can be found in appendix. Additionally, we present the results of quantitative

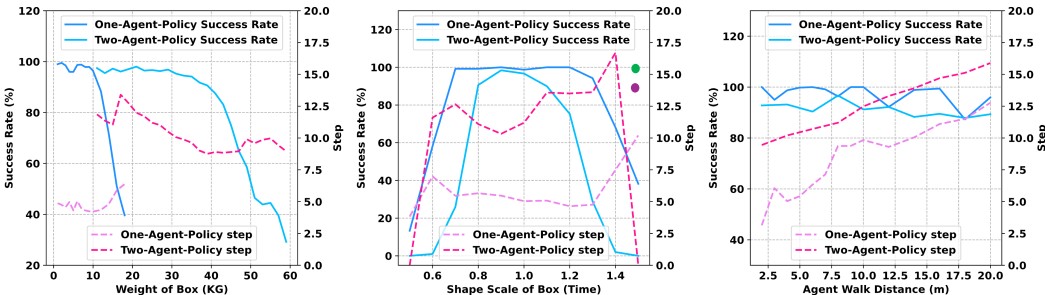

Figure 5: Detailed ablation experiments on single and two agents cases. "Step" measures the average consumed time in the successful cases. In the 2nd figure, the green circle represents the single-agent scenario without scaling the object's width, while the purple circle represents the multi-agent scenario.

experiments in Table 2, which indicate that our policy can effectively handle most daily life objects with a high success rate. Due to the greater variability in the shape and weight of sofas, the accuracy in Line 4 shows a slight drop compared to the result in Line 4 of Table 1. It is important to note that the significant variations in object shapes make it challenging to learn a unified object representation with a limited amount of data, especially since our input is merely a simple bounding box. Therefore, for single-person transport, different types of objects need to be trained separately from scratch, similar to how we handle boxes. For multi-person transport, objects like sofas, similar to boxes, require fine-tuning based on the weights obtained from single-person box transport.

## 4.3 Ablation Studies

To evaluate and understand the importance of different design choices in CooHOI, we conduct a detailed analysis of scenarios involving single and multiple agents. This includes boundary analysis, which explores factors that lead to a decrease in the success rate during object carrying. Furthermore, to fully validate the scalability of our method, we also tested and included the results in a four-person scenario. To ensure the reliability of our results, our experiments continue to utilize the average of 4 random seeds as previously mentioned.

**Single-Agent-Based Box Carrying.** We have demonstrated our framework's effectiveness in Table 1. Moreover, we aim to further investigate the maximum potential of our approach. We examined the robustness of the single-agent policy from various perspectives, including object shape, scale, weight, and trajectory length, as shown in Figure 5. First, we observed that a single agent can handle approximately 13 kg of weight. Continuously increasing the weight makes it difficult for the agent to lift the object. As mentioned in 4.2, since our policy input does not include object density, it is challenging for the agents to generalize across different weights. When the weight reaches 20 kg, the accuracy drops to just 30%. Additionally, due to the limited reach of human arms and the lack of dexterous hands in our agent, it is challenging for the agent to lift boxes that are either too large or too small. To validate this hypothesis, we restricted the object's width to 1x while scaling the length and height to 1.5x. As shown by the green circle in the second figure of Figure 5, the success rate could reach **97.8%**. Furthermore, our strategy is quite robust to distance variations and is generally unaffected by them.

**Multi-Agent-Based Box Carrying.** We conducted upper-limit testing on the multi-agent policy using a similar evaluation method as the single-agent case. As shown in Figure 5, the conclusions for two agents are similar to those for a single agent. The curves in the first figure demonstrate that with an increase in the number of agents, we can easily lift larger and heavier objects that a single person cannot handle, highlighting the necessity of having multiple agents. However, due to the increased complexity of coordinating two agents, boundary conditions have a more pronounced impact on the policy. For instance, objects that are excessively large or small can cause the policy to fail. As shown by the purple circle in the second figure of Figure 5, we conducted a similar experiment to the single-agent scenario. By restricting the object's width to 1x and scaling the length and height to 1.5x, the success rate increased from 0 to **88.67%**.

**Analysis of CooHOI Framework.** To thoroughly investigate the contribution of CooHOI, we analyzed the results of each method mentioned in Section 3 separately, as shown by the training curves in Figure 8. The first factor is the influence of the **Stand Point**, which refers to whether an extra point is introduced in front of the object to encourage the agent to walk toward it. During the experiments, we discovered that without this, the agent sometimes fails to approach the shortest edge of the object, resulting in a lower hold reward. This leads to an incomplete lift and the subsequent inability to carry the object. **Dynamic Observation** is the second factor, indicating whether we use dynamic information as observation for each agent. Without it, the observation for each agent is limited to the state information of the long object. We found that without the dynamics information, the agent just stands in front of the object, seemingly unsure of what to do. **Reverse Walk** indicates whether the training process includes motion data for walking backward and a reward function focused on learning this movement. We found that if the policy for a single agent is restricted to only forward movement, training with two agents then leads to a deadlock state. As shown in Figure 8, the agents might be able to contact the box, but they cannot carry it to the destination. **Initialization** refers to whether the two-agent policy is initialized using the single-agent policy and then be fine-tuned. In our experiment, even with extended training duration, training the two-agent policy from scratch still failed to achieve successful carrying, as shown in Figure 8. Based on the results above, the absence of any of the aforementioned methods causes our policy to fall into a locally suboptimal solution, preventing the completion of the transportation task. Moreover, you can also find more interesting visual examples of the above failure cases while extending agent numbers from one to two in the appendix.

## 5 Conclusion and Limitation

In this paper, we present Cooperative Human-Object Interaction (CooHOI), a framework designed to address cooperative object transporting tasks through a two-phase learning approach. By initially focusing on individual skill mastery via the Adversarial Motion Priors (AMP) method, followed by a strategic transition to multi-agent collaboration using Multi-Agent Proximal Policy Optimization (MAPPO), our approach facilitates a sophisticated interplay of shared dynamics and implicit communication among agents, resulting in an efficient and generalized performance.

However, within the scope of this paper, our huamnoid characters lack dexterous hands, which limit their ability to manipulate slippery objects or perform more precise actions. We also utilize object bounding box information as the goal feature for our task, which limits our framework's capacity to generalize to a diverse range of object shapes. Additionally, our experiments are limited to humanoid characters in simulation, without any real-world deployment. In future research, we aim to incorporate dexterous hands to enable the manipulation of a wider variety of objects, as well as integrate active perception and navigational abilities to make our framework more generalizable.

## Acknowledgments

This work is supported by Shanghai Artificial Intelligence Laboratory. Ziqin Wang is supported in part by Key R&D Program (2022ZD0115502), Natural Science Foundation (NO. 62122010, U23B2010), Zhejiang Provincial Natural Science Foundation under Grant No. LDT23F02022F02, Key Research and Development Program of Zhejiang Province under Grant 2022C01082, "Pioneer" and "Leading Goose" R&D Program (No. 2024C01161).

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

# Appendix

In this section, we categorize our discussion into three main parts. Initially, we delve into the sources and processing methods for motion data used in training. Following that, we explore how observations are constructed and how reward functions are established. Finally, we describe the implementation details including physics simulation and hyperparameters in network training.

## A    Sources and Processing of Motion Data

We collected a total of four types of basic reference motion data, including 9 motions related to walking, 5 related to picking up, 4 related to carrying, and 5 related to putting down. All these data are in SMPL format and recorded at 30 fps over 139 frames. They all originate from the ACCAD subset of the AMASS [17] dataset. Additionally, to ensure the stability of cooperative tasks involving multiple individuals, we included data for sidewalk and reverse carry motions. The sidewalk data comes from the CMU subset within AMASS, while reverse carry data was scarce. Therefore, we created reverse carry data by reversing the process of the carry data. In total, we used 26 motion data as references. Additionally, we performed a simple visualization of the extended objects as in Figure 6, which sampled from dataset [6].

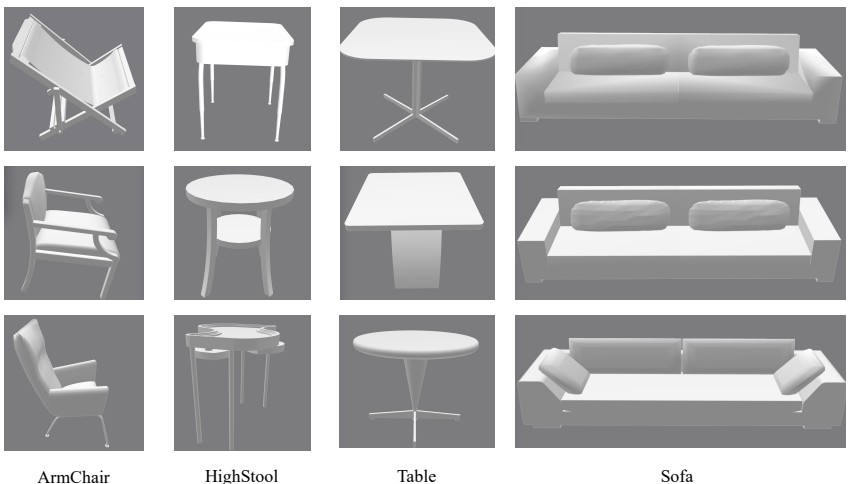

ArmChair          HighStool          Table          Sofa

Figure 6:   Some visualization of daily-life objects.

## B    Task Formulation

We formulate our approach as goal-conditioned reinforcement learning. At each discrete step $t$, the policy $\pi\left(a_t \mid \mathbf{s}_t, \mathbf{g}_t\right)$ generates an action $\mathbf{a}_t$, based on the current state $\mathbf{s}_t$ and a goal-specific feature $\mathbf{g}_t$. Following this action, the environment transitions into a subsequent state, and the agent receives a reward $r_t$. An episode concludes either after reaching a predetermined length or if conditions for early termination (ET) are met. Further details are provided below.

### B.1    Task Observation

The observational for the task is divided into two primary elements: the state feature $\mathbf{s}$, which encapsulates the character's bodily configuration, and the goal feature $\mathbf{g}$, which pertains to tasks involving object manipulation.

The state feature $\mathbf{s}$ is constituted by a 225-dimensional vector, encompassing:

- Height of the root: 1 dimension.

- Rotation of the root: 6 dimensions.
- Linear and angular velocity of the root: 6 dimensions.
- Position of local joints: 42 dimensions.
- Rotations of local joints: 84 dimensions.
- Linear and angular velocity of local joints: 84 dimensions.

While the root height is measured in the global reference frame, all other components are defined in the frame local to the character. Rotations follow a 6-dimensional representation for continuity [46]. The simulated character aligns with [25, 24, 7, 21], featuring 12 internally movable joints and a total of 28 degrees of freedom.

The goal feature $\mathbf{g}$ comprises a 75-dimensional vector, including:

- Position of the object: 3 dimensions.
- Rotation of the object: 6 dimensions.
- Dynamics of the object, which cover the bounding box position, along with linear and angular velocities: 33 dimensions.
- Target location: 3 dimensions.
- Target orientation: 6 dimensions.
- Dimensions of the target's bounding box: 24 dimensions.

These are measured in the frame local to the character.

## B.2 Reward Functions

The agent's reward $r_t$ at each time step $t$ is defined by

$$r_t = w^G r^G \left( \mathbf{s}_t, \mathbf{g}_t, \mathbf{s}_{t+1} \right) + w^S r^S \left( \mathbf{s}_t, \mathbf{s}_{t+1} \right) \tag{7}$$

Follow the formulation of the AMP framework [25], the **style reward** $r^S$ is calculated according to the discriminator:

$$r^S \left( \mathbf{s}_t, \mathbf{s}_{t+1} \right) = - \log \left( 1 - D \left( \mathbf{s}_t, \mathbf{s}_{t+1} \right) \right) \tag{8}$$

And the discriminator is trained by the following objective:

$$\begin{aligned}
\arg \min_D & - \mathbb{E}_{d^{\mathcal{M}}(\mathbf{s}, \mathbf{s}_{t+1})} \left[ \log \left( D \left( \mathbf{s}, \mathbf{s}_{t+1} \right) \right) \right] \\
& - \mathbb{E}_{d^{\pi}(\mathbf{s}, \mathbf{s}_{t+1})} \left[ \log \left( 1 - D \left( \mathbf{s}, \mathbf{s}_{t+1} \right) \right) \right] \\
& + w^{\text{gp}} \mathbb{E}_{d^{\mathcal{M}}(\mathbf{s}, \mathbf{s}_{t+1})} \left[ \left\| \nabla_\phi D(\phi) |_{\phi = (\mathbf{s}, \mathbf{s}_{t+1})} \right\|^2 \right]
\end{aligned} \tag{9}$$

The **task reward** function $r^G$ is generally segmented into three components, as in Equation (10): 1) $r^G_{\text{walk}}$, which encourages the agent to approach the object intended for manipulation. 2) $r^G_{\text{held}}$, which encourages the agent to align the center of its hands with the center of the box. 3) $r^G_{\text{target}}$, which encourages the agent to transport the object to the specified destination.

$$r^G = 0.2 * r^G_{\text{walk}} + 0.4 * r^G_{\text{held}} + 0.4 * r^G_{\text{target}} \tag{10}$$

The walk reward $r^G_{\text{walk}}$ is formulated as Equation (11), where $x_t^{\text{standing}}$ denotes the position of the standing point near the object, $v^*$ denotes the target velocity, and $d^*$ denotes the desired direction from root to the object.

$$r^G_{\text{walk}} = \begin{cases}
0.4 \exp \left( -0.5 \left\| x_t^{\text{standing}} - x_t^{\text{root}} \right\|^2 \right) + & \\
0.4 \exp \left( -2.0 \left\| v^* - d_t^{\text{root}} \cdot \dot{x}_t^{\text{root}} \right\|^2 \right) + & \\
0.2 \left\| d^* \cdot d_t^{\text{root}} \right\|^2, & \left\| x_t^* - x_t^{\text{root}} \right\| > 0.2m \\
1.0, & \text{otherwise}
\end{cases} \tag{11}$$

The held reward $r_{\text{held}}^{G}$ is formulated in Equation (12), where $x_t^{\text{hand}}$ denotes the center of the agent's two hands and $h_t$ is the position of the object helding point.

$$r_{\text{held}}^{G} = \exp\left(-5.0\|x_t^{\text{hand}} - h_t\|^2\right) \tag{12}$$

The target reward $r_{\text{target}}^{G}$ consist of two parts, $r_{\text{carry}}$ and $r_{\text{face}}$, as described in Equation (13).

$$r_{\text{target}}^{G} = 0.5 * r_{\text{carry}} + 0.5 * r_{\text{face}}. \tag{13}$$

The face reward $r_{\text{face}}$ guides the agent to walk either forwards or backward. As shown in Equation (14), this is achieved by comparing the agent's velocity direction with its orientation relative to the endpoint's location, thereby cultivating the agent's proficiency in bidirectional locomotion.

$$r_{\text{face}} = \begin{cases} x_t^{\text{face}} \cdot v_t^{\text{face}}, & x_t^{\text{face}} \cdot (d_t - x_t^{\text{root}}) \geq 0 \\ -x_t^{\text{face}} \cdot v_t^{\text{face}}, & x_t^{\text{face}} \cdot (x_t^{\text{root}} - d_t) \geq 0 \end{cases} \tag{14}$$

The carry reward $r_{\text{carry}}$, is designed to guarantee that the object is delivered to the precise location at a specific angle. As outlined in Eq. 15, we constrain the agent's movement direction, alongside the proximity to the end destination and the intended angle. Within this context, $x_t^*$ signifies the 3D coordinates of the destination, while $p_t^*$ represents the 2D destination coordinates. Similarly, $p_t^{\text{root}}$ indicates the 3D position of the agent's root. Furthermore, $\text{rot}^*$ designates the object's desired orientation.

$$r_{\text{carry}} = \begin{cases} 0.5 * r_t^{\text{near}} + 0.25 * r_t^{far} + 0.25 * r_t^{\text{dir}}, & \|x_t^* - x_t^{\text{root}}\| > 0.1m \\ 0.5 * r_t^{\text{near}} + 0.25 * r_t^{\text{dir}} + 0.25, & \text{otherwise}, \end{cases} \tag{15}$$

where

$$r_t^{\text{far}} = \exp\left(-0.5\left\|p_t^* - p_t^{\text{root}}\right\|^2\right)$$

$$r_t^{\text{near}} = \exp\left(-10.0\left\|x_t^* - x_t^{\text{root}}\right\|^2\right)$$

$$r_t^{\text{dir}} = \left\|\text{rot}^* \cdot \text{rot}_t^{\text{object}}\right\|^2$$

### B.3 Reset and early termination condition

An episode ends either after reaching a predetermined duration or upon the activation of early termination (ET) conditions. During our experiments, we observed that lower object heights could lead to kicking actions, where the agent tend to kick the object to destination, significantly slowing down the training process. To address this, we assess the object's velocity and height to determine the presence of kicking phenomena. If the height of the object is lower than 0.3m and its velocity in x-y plane is greater than 1m/s, the kicking early termination (KET) condition is triggered. Experimental results show that this strategy significantly stabilize the training process.

## C   Implementation Details

### C.1   Training Details.

Adopting the methodology of AMP [25], we develop a low-level controller encompassing both policy and discriminator networks. The policy network is bifurcated into a critic and an actor-network, each initiating with a CNN layer and proceeding to two MLP layers configured with [1024, 1024, 512] units. The discriminator network is similarly structured, featuring two MLP layers with [1024, 1024, 512] units. We select PPO [28] as the primary reinforcement learning algorithm, coupled with the Adam optimizer [12] at a learning rate of 2e-5. The only difference between the multi-agent setting and the single-agent setting during training is whether a pre-trained weight is loaded. Our experiments are conducted on the IsaacGym simulator [18] using a single Nvidia GTX 3090Ti GPU. We run 4096 parallel environments across 15,000 epochs, which takes approximately 15 hours to complete.

## C.2 Hyperparameters

Following previous work[25, 7, 21], we use the Isaac Gym simulator [18]. The simulation runs at 60Hz and the control policy runs at 30Hz.

Besides, the hyperparameters we used in the training process is detailed below:

Table 3: Hyperparameters for CooHOI.

| Parameter | Value |
| --- | --- |
| Number of Environments | 4096 |
| $w_G$ Task-Reward Weight | 0.5 |
| $w_S$ Style-Reward Weight | 0.5 |
| PPO Minibatch Size | 16384 |
| AMP Minibatch Size | 4096 |
| Horizon Length | 32 |
| Learning Rate | $2e-5$ |
| PPO Clip Threshold $\epsilon$ | 0.2 |
| $\gamma$ Discount | 0.99 |
| GAE ($\lambda$) | 0.95 |
| $T$ Episode Length | 600 |

## D   More Ablation Studies and Visulizations.

**Failure case visualization.**   Here, we conducted a visual analysis of the fail cases. First, for the case lacking a stand point, we can clearly see that the agent moves towards the nearest face, even though it is not the shortest edge, which leads to the agent's inability to carry the object. In the second image, in the absence of dynamic input, we observe that the agent stands still, unable even to squat. In the third image, which depicts the scenario without reverse walking, the agent is able to lift the box, but because it cannot learn the backward gait, the two agents end up pushing the box against each other, causing a deadlock.

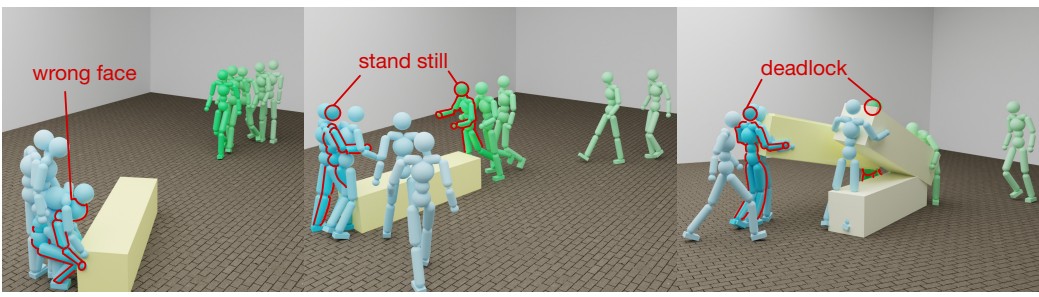

Figure 7:  Some visualization on failure cases. "Stand point" means a leading point behind the object to encourage the agent to walk to the object. "Dynamic Observation" means that each agent has its unique input. "Reverse Walk" indicates whether a single agent possesses the skill to walk backward. Without any of the methods we propose, the policy cannot be successfully trained.

**Ablation on performance of CooHOI under noisy scenarios.**   In CooHOI, all state information is provided using ground truth data from simulators. However, in real-world settings, input data is often noisy and prone to errors. To evaluate the robustness of our framework in such conditions, we introduce random noise into the observation space of our policy and assess its performance under observation noise, as shown in Table 4.

Table 4: Results of our policy under noisy conditions: We tested both single-agent and two-agent box-carrying scenarios. The noise level is defined by the standard deviation of the Gaussian noise used. SR stands for success rate. The definitions of success rate and precision are consistent with those in Section 4.1 of our paper.

| Agent # | Weight(kg) | Noise | SR (%) | Precision (cm) |
|---|---|---|---|---|
| 1 | 10 | 0 | 96.85 | 5.76 |
| 1 | 10 | 1 | 95.80 | 6.82 |
| 1 | 10 | 2 | 78.56 | 9.28 |
| 1 | 10 | 3 | 60.03 | 10.75 |
| 1 | 10 | 4 | 48.48 | 8.62 |
| 2 | 20 | 0 | 90.33 | 8.80 |
| 2 | 20 | 1 | 90.23 | 8.96 |
| 2 | 20 | 2 | 87.98 | 8.92 |
| 2 | 20 | 3 | 84.86 | 9.01 |
| 2 | 20 | 4 | 79.93 | 9.28 |

**Training reward curves.** To enhance visualization for ablation studies, we plot the training curves for both the carry reward and held reward in the two-agent training setup. The results, shown in Fig 8, illustrate the effectiveness of our CooHOI framework.

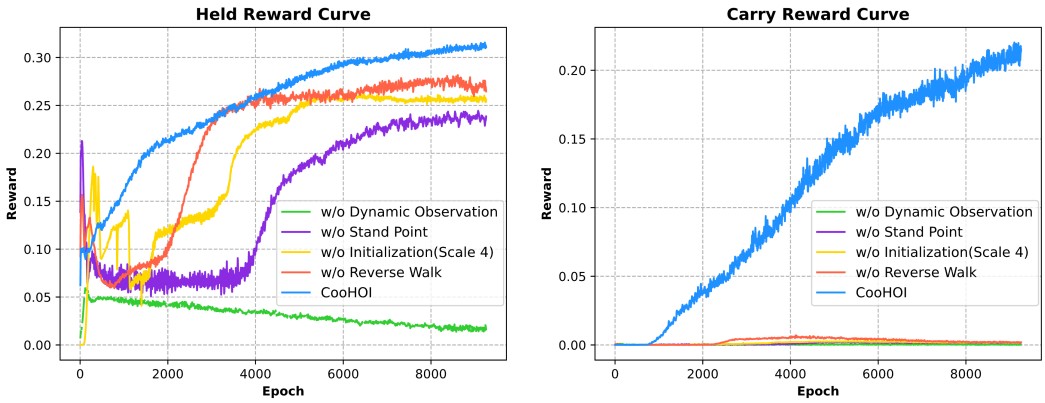

Figure 8: Training curves for carry reward and held reward in the two-agent setting, using four random seeds, consistent with the definitions provided in Section 3. To ensure different models were trained for the same duration, we extended the training steps for the 'From Scratch' model by a factor of 4, as indicated by 'Scale 4' in the graph. The curves were plotted by sampling every four frames.

