# OpenReview forum: "CooHOI: Learning Cooperative Human-Object Interaction with Manipulated Object Dynamics"
_NeurIPS.cc/2024/Conference — NeurIPS 2024 spotlight_

### Official Review · Reviewer_TU4S · 2024-07-04

**Soundness:** 4
**Presentation:** 4
**Contribution:** 4
**Rating:** 7
**Confidence:** 5

**Summary:**

This paper works on learning multi-human cooperative object manipulation, specifically the collaborative carrying of objects.
This paper proposes a two-stage method to learn collaborative object carrying. In the first stage, the agent learns single-person object carrying from motion capture data and heuristic-based task rewards following the Adversarial Motion Priors (AMP) framework. In the second stage, the agent policy is fine-tuned using multi-agent reinforcement learning to learn multi-agent collaborative object carrying.
Experiments show that the trained policy can successfully control the agents to complete collaborative objects carrying tasks given objects of varying shapes and weights.

**Strengths:**

1. The proposed cooperative interaction learning framework can learn multi-human collaborative behaviors requiring only single-human motion capture data. In addition, the learned policy has better generalization compared to tracking-based methods.

2. The implicit communication via object dynamics unifies the observation in the single-human and multi-human stages, reducing the gap between the two stages and facilitating the learning of collaborative object carrying.

3. This paper presented extensive ablation study and boundary analysis to show how and to what extent the proposed method works.

**Weaknesses:**

1. This paper simulates an oversimplified humanoid without hand modeling considering the fact that it works on object manipulation problems, for which the hands play essential roles. The authors also recognize this issue leads to capacity limitations.

2. The agent observation features may not be sufficient for more complex manipulations. In the presented paper, the agent observation mainly consists of the self-motion and cropped object bounding box features. However, more complex collaborative tasks like carrying articulated objects and assembling a sofa require a global overview of the object instead of only a local observation. In addition, the agent also needs to observe the collaborators to avoid interference such as grasping at the same point.

3. The proposed method needs to train a separate policy for a category of objects with similar shapes (L246-247), which indicates a generalization problem.

**Questions:**

1. How the object weights can affect the humanoid in the simulation? Is there an upper bound for the joint force of the simulated humanoid? How does the humanoid apply forces to the object in the simulation?

2. Are the standing point and held point also part of the goal inputs? If not, during the training, the same motion sequence can have different rewards given differently sampled standing points? How are the standing points sampled given an arbitrary object mesh?

**Limitations:**

The author discussed the limitation of lacking dexterous hands, which limits the agent interaction capacity and is left as future work.
Please also refer to the weakness discussion above.

---

> ### Author Rebuttal · Authors · 2024-08-07
>
> Thank you for your valuable comments and feedback. We hope the following clarifications address your concerns.
>
> > W1: This paper simulates an oversimplified humanoid without hand modeling considering the fact that it works on object manipulation problems, for which the hands play essential roles. The authors also recognize this issue leads to capacity limitations.
>
> This is indeed a limitation of our work. Our framework currently handles 40 different common mid-size daily life objects, but incorporating dexterous hands would definitely enable humanoids to interact with small objects or objects with unusual shapes. This is a future direction for our research.
>
> > W2: The agent observation features may not be sufficient for more complex manipulations. In the presented paper, the agent observation mainly consists of the self-motion and cropped object bounding box features. However, more complex collaborative tasks like carrying articulated objects and assembling a sofa require a global overview of the object instead of only a local observation. In addition, the agent also needs to observe the collaborators to avoid interference such as grasping at the same point.
>
> In our work, we minimize interference between different agents by utilizing designs such as standing points and held points. These designs have proven effective for multi-agent object transportation tasks.
>
> However, for more complex tasks like carrying articulated objects or assembling a sofa, additional observation input is indeed necessary. These tasks may require explicit observation and communication between agents, as well as specialized designs.
>
> > W3: The proposed method needs to train a separate policy for a category of objects with similar shapes (L246-247), which indicates a generalization problem.
>
> Yes, we train a separate policy for each category of objects with similar shapes. Our main focus is on training multi-agent cooperative object transportation, so we did not introduce special designs for different shapes and categories. This approach allows us to concentrate on the primary contribution of our work.
>
> Also, we think that handling different object meshes is primarily a perception problem, whereas our focus is on the control aspect of cooperative HOI tasks. Incorporating object mesh-related information or other perception-related designs would certainly enhance the generalizability of our method. This will be a future direction of our study.
>
> > Q1: How the object weights can affect the humanoid in the simulation? Is there an upper bound for the joint force of the simulated humanoid? How does the humanoid apply forces to the object in the simulation?
>
> In our experiments, we use the same humanoid model as previous work[1][3][5][6][7][11]. To reiterate, the action $a \in \mathbb{R}^{28}$ specifies the target positions for PD controllers at each joint, and the forces are calculated as `(target_pos - dof_pos) * stiffness - dof_vel * damping`.
>
> - We are not entirely sure what you mean by “How do object weights affect the humanoid in the simulation?”. Perhaps you are asking about the forces at each joint when carrying objects of different weights? We have visualized this in Figure3 and Figure4 in the PDF in the global rebuttal.
> - Yes, there is an upper bound for the joint force of the simulated humanoid. Specifically, since we are using positional control, every degree of freedom (dof) has limits, namely, dof_limits_lower and dof_limits_upper, and the resulting force is bounded.
> - When the humanoid manipulates objects, there will be friction forces between the hands of humanoid and the objects, which are calculated by the PhysX phyics engine of Isaac Gym.
>
> > Q2: Are the standing point and held point also part of the goal inputs? If not, during the training, the same motion sequence can have different rewards given differently sampled standing points? How are the standing points sampled given an arbitrary object mesh?
>
> We apologize for any confusion. In our work, the standing points are part of the goal input, while the held points are not. During single-agent training, the standing points are **randomly sampled** to positions directly in front of various faces of the object. During multi-agent training, the standing points are sampled in front of each end of the long object. The held points are not part of the goal inputs because we set them at the geometric center of the object, allowing the agent to “infer” the position of the held point based on the object’s position and its bounding box information. We will briefly introduce the constitution of goal feature in Section 3.2.1 in the next version of our paper.
>
> If the standing points and the object’s position and rotation are given, then the same motion sequence will yield the same rewards.

---

> ### Comment · Reviewer_TU4S · 2024-08-08
>
> I appreciate the additional explanation and visualization provided in the rebuttal. However, the H-shape box example does not solve the concerns about generalization because it needs a specially designed object shape for the 4-person cooperation to work.
> Generalization capability to various objects and interaction skills remain a weakness of this work. Moreover, simulating a humanoid without dexterous hand modeling inherently limits the method to work for interactions requiring complex controls, as also raised by other reviewers.
>
> Overall, I still appreciate the contributions of exploring learning cooperative human-object interaction, the design of object dynamics-based state observation and collaborative training framework, and extensive experiments. I believe this paper makes a valuable contribution and recommend accepting it.

---

> > ### Author Response · Authors · 2024-08-09
> >
> > Thank you for your insights and suggestions. The H-shape box represents our effort to explore the boundaries of CooHOI’s generalization capability regarding the number of agents. In future work, we will continue to exploring its generalization capability across various objects and interaction skills, incorporating more experiments and settings involving dexterous hands.

---

### Official Review · Reviewer_kufk · 2024-07-12

**Soundness:** 3
**Presentation:** 3
**Contribution:** 3
**Rating:** 6
**Confidence:** 4

**Summary:**

This work address the problem of multi-character collaboration for object transporting tasks. Different from previous works approaching the multi-character HOI task with tracking-based method, this work learns a physics-based multi-agent policy with reinforcement learning. Instead of directly training a multi-agent policy from scratch, the authors suggest using a two-phase learning approach: CooHOI uses a two-phase learning approach: individual skill acquisition through Adversarial Motion Priors (AMP) and then collaborative learning with Multi-Agent Proximal Policy Optimization (MAPPO). Agents learn to coordinate implicitly by responding to changes in object dynamics caused by others.

**Strengths:**

* The manuscript is well-written, the methodology is clearly presented and easy to follow.
    * This paper is addressing an under-studied and important application - the physics-based multi-agent cooperation HOI, where most previous works explores in the single-agent setup.
    * The proposed two-state learning pipeline is reasonable and few important designs including the stand point, held point shows to be effective in addressing this multi-agent cooperation tasks.
    * The ablation study and analysis provide comprehensive analysis and capacity boundary test for the proposed pipeline and these provide useful insights for the community.

**Weaknesses:**

* This paper build a framework for multi-agent cooperation policy learning with AMP for single-agent policy pertaining and MAPPO for cooperation skill learning. The technical contribution to the community is minor.
* The method include few heuristic designs to simplify the problem, for example, the manually defined held point, and object bounding box observation, making it not easy to generalize and scale to more diverse objects.
* The current framework can be hardly generalise to objects with different size and different shape, and the human motion styles are quite limited.
        * Regarding generalization to objects weights and scales, are the results presented in Figure 5  the transferring results to different weights and scales or the per-object training results.
        * Regarding the motion styles, in the training dataset, limited motion sequences are involved with only walking, lifting up and putting down, would the proposed method build upon AMP be able to cover more diverse skills, and this could be helpful to generate more natural and diverse HOI motions.
* It is presented that when increasing the number of characters, the policy has problem in putting the object down due to the absence of dexterous hands and limited friction. Could the author elaborate more on the major bottlenecks encountered when dealing with a larger number of characters? Assuming that the friction parameter can be adjusted in the simulator and should not be a significant issue for motion generation tasks purely within the simulation, what specific contributions could dexterous hands make to this scenario? Additionally, while the introduction of dexterous hands would undoubtedly enhance the agent's capabilities, what other factors should be considered?

**Questions:**

See more in the weakness section :-)

**Limitations:**

Yes, the authors addressed the limitation.

---

> ### Author Rebuttal · Authors · 2024-08-07
>
> Thank you for your insightful comments. Many of the weaknesses you pointed out are exactly the areas we plan to address in future work, as we aim to make our method more generalizable.
>
> > Q1: This paper builds a framework for multi-agent cooperation policy learning with AMP for single-agent policy pertaining and MAPPO for cooperation skill learning. The technical contribution to the community is minor.
>
> We acknowledge that the AMP and MAPPO algorithms we used are not new to the community. However, we believe our contribution lies in **presenting a proof-of-concept framework for this important and relatively under-studied subfield**: physics-based multi-agent cooperation in HOI. Our **core insight** is that manipulated object dynamics can serve as feedback in single-agent skill training, an implicit communication channel in multi-agent coordination, and an interface to facilitate efficient skill transfer from single to multi-agent scenarios. We believe this will inspire many in the kinematics-based animation, physics-based animation, and robotics communities.
>
> Additionally, some previous tracking-based efforts [18] on physics-based multi-agent cooperation in HOI have addressed this issue by utilizing **high-quality multi-agent mocap data**. Their methods cannot train successfully without multi-agent mocap data, as simply using AMP (or other types of style rewards) and MAPPO would be **too challenging**. This is demonstrated by the "w/o CooHOI" result in Table 1 and the "w/o Initialization" in Figure 6. In comparison, our framework only requires single-agent motion capture data and can extend to different types of objects and varying numbers of agents by utilizing the manipulated object dynamics.
>
> > Q2: The method includes a few heuristic designs to simplify the problem, for example, the manually defined held point, and object bounding box observation, making it not easy to generalize and scale to more diverse objects.
>
> Our work primarily focuses on multi-person object transportation tasks, validating that using manipulated object dynamics as an implicit communication channel can efficiently facilitate skill transfer and coordination learning. The designs in our methods are mainly intended for efficient skill transfer from single-agent skills to multi-agent coordination learning. We **did not introduce special designs for objects with different shapes, but it can still generalize to 40 different shapes of objects**.
>
> We believe that handling different object meshes is primarily a perception problem, whereas our focus is on the control aspect of cooperative HOI tasks. Incorporating object mesh-related information or other perception-related designs would certainly enhance the generalizability of our method. This will be a future direction of our study.
>
> > Q3: The current framework can be hardly generalize to objects with different size and different shape, and the human motion styles are quite limited.
>
> - Regarding object weights and scales, the results in Figure 5 are the **transfer results to different weights and scales**. This is a **boundary analysis** of our framework: we first train agents to carry objects with different weights and scales, and then test when they fail with out-of-distribution scales and weights. The results indicate that the policy performs well with out-of-distribution scales and weights but eventually fails when these parameters deviate too far from the training conditions.
> - Regarding object shapes, our framework can generalize to 40 different common mid-size daily-life objects. We have thoroughly discussed this in our answer to your question Q2.
> - Regarding generalization to motion styles, since our work mainly focuses on multi-agent cooperative object transportation, we only chose walking, lifting up and putting down objects, and reverse walking to cover the motion styles.
>
> Your feedback on the generalizability of our methods for both object shapes and motion styles is invaluable to us. We plan to enhance our framework to address **more diverse cooperative HOI tasks in future work**. This is also discussed in our limitations section. After incorporating dexterous hands, we will work on enabling the humanoid characters to cooperatively handle objects of different sizes using object mesh-related information in our training pipeline and aim to achieve various motion styles by utilizing motion priors or motion latents.
>
> > Q4: It is presented that when increasing the number of characters, the policy has problem in putting the object down due to the absence of dexterous hands and limited friction. Could the author elaborate more on the major bottlenecks encountered when dealing with a larger number of characters?
>
> - We attribute this failure to the absence of dexterous hands and limited friction because, during the process of carrying the box to the destination, the large box sometimes slips from the humanoid agent’s hands because the hand model is spherical, making it very difficult to hold the corners of the box, which tends to be “squeezed out” of their hands. However, we found a way to bypass this limitation without dexterous hands: we had the four agents carry an “H-shaped” large box and successfully complete the task. **Please refer to the global rebuttal for more details.**
>
> - Regarding the major bottlenecks encountered with a larger number of characters (more than 4), we believe the main challenge lies in ensuring that these agents can effectively communicate with each other using object dynamics. The complexity of this problem increases exponentially as the number of agents grows.
>
> - We believe introducing dexterous hands would allow the agents to handle smaller daily-life objects or objects with unusual shapes, such as two agents cooperatively carrying a TV together.

---

> > ### Comment · Reviewer_kufk · 2024-08-12
> >
> > Thanks a lot for the authors' detailed response and additional experiments and visualization. The additional experiment on the H-shape objects transporting with 4 agents demonstrates the necessity of carefully selected object types, while I agree that generalizing to challenging object shapes remains a very challenging task, and the author also gives insights on how to further approach it.
> >
> > Overall, I appreciate the efforts in this work to build a framework to tackle the multi-agent HOI problem, and I have also read other reviews as well as authors' responses, and I will maintain my original rating as weak accept.

---

> > > ### Author Response · Authors · 2024-08-13
> > >
> > > Thanks for your insights and valuable feedback!

---

> ### Comment · Area_Chair_HGdc · 2024-08-12
> **To reviewer kufk : Please respond to rebuttal**
>
> Hi reviewer kufk ,
>
> Thank you for your initial review. Please kindly respond to the rebuttal posted by the authors.
> Does the rebuttal answer your questions/concerns? If not, why?
>
> Best,
> AC

---

### Official Review · Reviewer_Zw8u · 2024-07-13

**Soundness:** 4
**Presentation:** 3
**Contribution:** 3
**Rating:** 7
**Confidence:** 3

**Summary:**

In this paper, the authors introduce a novel framework, Cooperative Human-Object Interaction (CooHOI), aimed at tackling the problem of multi-agent object transportation. The framework consists of two phases: initially, a single agent learns to perform tasks, followed by multiple agents learning to collaborate through shared dynamics of manipulated objects. The utilization of shared object dynamics has proven effective in learning cooperative tasks. The authors conduct comprehensive experiments to validate the efficacy of their approach and explore its capability boundaries.

**Strengths:**

* The idea of sharing manipulated object dynamics among multiple agents as implicit communication to facilitate multi-agent collaboration, is intuitively appealing and has proven effective. This approach aligns with object-centric concept, designed to leverage object dynamics to guide agent actions and promote collaboration. It provides valuable insights for future research.
* The authors conduct extensive experiments to validate the effectiveness of their approach, including many ablation studies to assess the impact of different design choices, as well as exploration into the framework's capability boundaries.
* The paper is well-organized and easy to understand.
* The inclusion of diverse visualizations enhances qualitative comprehension of the method's performance.

**Weaknesses:**

* It appears that the shapes and categories of the manipulated objects are not very diverse, potentially constraining the method's ability to generalize to novel objects. I recommend that the authors summarize the counts of shapes and categories used in the experiments.  Additionally, if the shapes of the manipulated objects are limited and the grasp actions are abstracted in the simulation, the authors are encouraged to compare CooHOI with some heuristic methods. An intuitive heuristic baseline could involve having two agents grasp each end of a long object, and then move at the same speed to the target position.
* While the authors provide detailed explanations in the appendix, it would enhance comprehension if they provide simple explaination when introducing a new concept in the main paper. Here are some examples:
  * In Section 3.1, the authors introduce the task-specific goal feature $g_t$ without explanation.
  * While it is understandable that the proposed style reward can help the method produce behaviors similar to those in the dataset, it would be better to provide a simple explanation of how to evaluate such similarity.
  * It would be beneficial to provide a brief introduction to the baseline InterPhys in one or a few sentences.

**Questions:**

* While acknowledging the idea of utilizing the object’s dynamics as implicit communication between agents, such communication is not synchronous. Specifically, agents adjust their actions only after one agent changes the object dynamics, resulting in a delay between the change in object dynamics and the response actions taken. I wonder if this delay could lead to jitter of the manipulated object. If not, could the reviewers discuss why such jitter would not occur?
* The held point is described as the geometric center of each end of the object. However, in Figure 1, when cooperatively carrying a large box, the characters grasp the edges of the box instead of the center of its face. I am unsure if I misunderstood the definition of the held point, or if the current definition is inadequate, leading agents to choose other positions they find more suitable for completing the task.

* In the second row of Figure 4, I am curious about why the four agents fail to put down the box. The authors attribute this failure to the absence of dexterous hands and limited friction, however, similar limitations exist in two-agent manipulation tasks. Besides, the multi-agent put-down operation closely resembles the multi-agent pick-up operation (but in reverse), while the agents can successfully pick up the box, they struggle to put it down. Could the authors provide more discussion or visualizations for better understanding?

**Limitations:**

The authors have provided a detailed discussion of limitation and failure cases.

---

> ### Author Rebuttal · Authors · 2024-08-07
>
> Thank you for your valuable comments and feedback.
>
> > W1.1 : It appears that the shapes and categories of manipulated objects are not very diverse, potentially constraining the method's ability to generalize to novel objects.
>
> We sampled nearly 40 common everyday objects for training, which fall into six categories: box, table, armchair, and high stool for single agents, and long box and sofa for two agents. We will summarize the counts of shapes and categories of the objects we used in Section 4.1, “Datasets and Initialization,” in the next version of our paper.
>
> In addition, the main focus of our work is on training multi-agent cooperative object transportation, so we did not introduce special designs for different shapes and categories. This allows us to concentrate on the main contribution of our work. Through fine-tuning, our framework can successfully enable single and multi-agent transportation of **common mid-size daily-life objects** (Table 2). As a comparison, [3] used 40 object shapes, [17] used 17 object shapes. [1] used 350 object shapes, although most of these shapes were for humanoids to sit or lie on. For carrying tasks, [1] primarily used the simple box shape. Therefore, we believe that the 40 shapes and 6 categories of objects we used are **sufficient to validate our framework**.
>
> > W1.2: The authors are encouraged to compare CooHOI with some heuristic methods. An intuitive heuristic baseline could involve having two agents grasp each end of a long object, and then move at the same speed to the target position.
>
> We believe this intuitive heuristic baseline is quite similar to the “w/o Dynamic Observation” scenario described in Section 4.3 and shown in Figure 6. The key difference between this baseline and our CooHOI method is that the baseline does not use dynamics information from each end of the object to facilitate skill transfer. Training multi-agent cooperation using only reward training is challenging because coordination between agents cannot be easily hard-coded.
>
> > W2: While the authors provide detailed explanations in the appendix, it would enhance comprehension if they provide simple explaination when introducing a new concept in the main paper.
>
> We will incorporate your suggestions in the next version of our paper to make it easier for readers to follow. Thanks for your suggestion!
>
> > Q1: I wonder if this delay of communication could lead to jitter of the manipulated object. If not, could the reviewers discuss why such jitter would not occur?
>
> In the evaluation of the trained policies, we did not observe significant jittering of the manipulated object. You can **see the visualization of our results in the “DemoVideo_CooHOI.mp4” included in our supplementary materials**.
>
> We believe this is due to the following reasons:
> 1. We model the objects as rigid bodies, so changes in object dynamics occur “instantly,” without causing large delays.
> 2. Our framework is similar to the “object-centric” concept, where object jittering would result in humanoid motion jittering. Since we use AMP to provide a “style reward,” this motion jittering is not encouraged, thus forcing the humanoid to learn effective cooperation and avoid object jittering.
> 2. We train the multi-agent policy in a cooperative setting, so implicit communication helps achieve coordination learning by maximizing the reward and avoiding object jittering since jittering does not maximize the reward.
>
> We believe that explicit or implicit communication in multi-agent systems should carefully address latency issues, especially in **real-world object-carrying tasks**. Thank you again for your valuable comment.
>
> > Q2: The held point is described as the geometric center of each end of the object. However, in Figure 1, when cooperatively carrying a large box, the characters grasp the edges of the box instead of the center of its face.
>
> The held point is defined as the **3D geometric center** of each end of the objects. We used the held point and the corresponding $r_{held}$ to encourage the agents to reach out their hands near the object and learn to manipulate it, where $r_{held}$  encourage the humanoid to put the center of its 2 hands closer to the held point.
>
> The held point design in our multi-agent HOI setting serves as "abstract indicators" for primarily distinguishing the guidance for each agent during the interaction with the same object, rather than specifying the exact position for them to place their hands. The precise position varies for each object shape, and we encourage the agents to learn this during training. We believe this is why our method can generalize to different object shapes without requiring their mesh information as input.
>
> > Q3:I am curious about why the four agents fail to put down the box.
>
> We attribute this failure to the absence of dexterous hands and limited friction. During the process of carrying the box to its destination, we found that the large box sometimes slips from the humanoid agents’ hands. This is because the hand model of the humanoid agents is **spherical**, making it very difficult to hold the corners of the box, causing it to be **“squeezed out” of their hands**.
>
> The difference between the four-agent scenario and the two-agent tasks is that, in the four-agent scenario, the humanoids can **only grasp the corners of the box**. Additionally, when the four humanoids are picking up the object, the two humanoids positioned diagonally can push forward against each other to prevent the box from slipping out of their hands. However, this is very difficult during transportation and often leads to the box constantly falling. As a result, they **never have the opportunity to learn the “putdown” action**, causing the entire process to fail.
>
> However, we found a way to bypass this limitation without dexterous hands: we had the four agents carry an “H-shaped” large box and successfully complete the task. **Please refer to the global rebuttal for more details**.

---

> > ### Comment · Reviewer_Zw8u · 2024-08-10
> >
> > Thank you for the clarification provided in the rebuttal. I will maintain my score and continue to support the acceptance of this paper.

---

> > > ### Author Response · Authors · 2024-08-13
> > >
> > > Thanks for your insight and valuable feedback!

---

### Official Review · Reviewer_ehnN · 2024-07-17

**Soundness:** 3
**Presentation:** 3
**Contribution:** 2
**Rating:** 6
**Confidence:** 4

**Summary:**

This paper proposes a framework for multi-agent cooperative manipulation in the context of humanoids carrying and transporting large furniture. This task is decomposed into several steps. First, they train a single humanoid to learn how to hold and carry relatively small objects. The agent is trained with ground-truth object information and AMP for natural humanoid behavior. Next, they train a multi-agent policy initialized from the single-agent policy for transporting larger objects. The authors present several ablations and report high performance compared to baseline methods in simulation.

**Strengths:**

This paper studies a practical task and solves it with a well-designed engineering system. It is also well-written, making it easy to read.

The paper presents comprehensive ablation experiments on both the single-agent and multi-agent policy parts, revealing important information for the framework.

The framework can learn natural humanoid behavior.

**Weaknesses:**

I’m curious about the robustness of the proposed system. If I understand correctly, the bounding box information is given as ground-truth parameters. However, in reality, these quantities are far from perfect.

The task is restricted to moving and transporting large furniture-type objects.

**Questions:**

In Table 1, what does the “w/o CooHOI” mean? Does that mean training the entire policy from scratch? Is that policy trained with AMP and the same rewards? And what is the definition of CooHOI? What is included in CooHOI?

Regarding the robustness of the system: How would the policy perform in noisy situations? For example, when the object bounding boxes are not accurate (in real scenarios, object perception will have errors).

In Figure 6, can you elaborate more on why policies, except for CooHOI, fail to carry objects even when they can still hold objects? Specifically, what are the failure cases? Do these policies fail to maintain a stable holding motion, or do they fail to accurately move the objects to the target location?

In addition, r_carry is only defined in the supplementary material. I suggest including it in Section 3.2.2.

It would be better to plot the standard deviations in Figure 6, given that you evaluate four seeds.

What is the action space of the policy? Is it the target joint position for each joint?

**Limitations:**

The authors provide a comprehensive limitation section of their method.

---

> ### Author Rebuttal · Authors · 2024-08-06
>
> Thank you for your valuable time and insightful comments. We hope the following clarifications address your concerns.
>
> > W1: I’m curious about the robustness of the proposed system. If I understand correctly, the bounding box information is given as ground-truth parameters. However, in reality, these quantities are far from perfect.
>
> Thank you for pointing this out. Yes, in our work, the bounding box information is provided as ground-truth parameters. While this may seem “unrealistic” in the real world, it is a **common setting** adopted by the physics-based character animation community [2][3][4][5][6][7][8][9][10][11][12][13], including the baseline method [1] we compared in our study. Additionally, the **main focus of our work is on training cooperative human-object interaction tasks**, such as carrying everyday objects. Therefore, we did not introduce special designs for noisy scenarios that may occur in real-world object detection. We believe this allows us to concentrate on the primary contribution of our work.
>
> > W2: The task is restricted to moving and transporting large furniture-type objects.
>
> Our method can handle 40 different shapes of common daily-life furniture-like objects. However, as we mentioned in the limitations section, incorporating dexterous hands would make our framework more generalizable to smaller and unusually shaped objects found in daily life. This is a direction of our future work.
>
> > Q1: In Table 1, what does the “w/o CooHOI” mean? Does that mean training the entire policy from scratch? Is that policy trained with AMP and the same rewards? And what is the definition of CooHOI? What is included in CooHOI?
>
> We apologize for the confusion regarding the term “w/o CooHOI” in Table 1. As stated in lines 233-235, “Without using our CooHOI framework and simply employing parallel training for multi-agent tasks, the training fails.” This refers to training the entire multi-agent policy with AMP and the same rewards from scratch.
>
> The definition of CooHOI **encompasses the entire framework design**. Specifically, it involves the insight that object dynamics information is crucial for both single-agent skill learning and coordination learning. The “bounding box” design serves as an interface for efficiently transferring single-agent skills to cooperative learning.
>
> In contrast, “w/o CooHOI” means not using object dynamics information when training multi-agent cooperation skills. This has two conditions:
> 1. Directly train the whole policy from scratch, with agents observing the box as is, while the AMP reward and the reward function remain the same. This corresponds to the “w/o Initialization” curve in Figure 6.
> 2. First training the single-agent skills, then fine-tuning using MAPPO, but in the multi-agent training stage, not utilizing object dynamics (the state of the bounding box at each side) and directly including the object state as the goal feature. This scenario is illustrated in Figure 6 and Appendix D, Figure 2, “No Dynamics Observation.” This results in agents standing near the object and seemingly forgetting how to interact with it.
>
> Thank you for pointing out this unclear explanation. We will clarify it in the paper’s next version.
>
> > Q2: How would the policy perform in noisy situations?.
>
> Although we didn't account for noise situations or introduce special designs for robustness training under noise conditions, we can still test our policy's performance. Specifically, we added Gaussian noise to the object dynamics information and varied the noise level by adjusting the standard deviation of the noise. A noise level of 1 indicates that the standard deviation of the noise is 1 cm. **Please refer to Table 1 in the PDF included in the global rebuttal for our results**.
>
> > Q3: In Figure 6, can you elaborate more on why policies, except for CooHOI, fail to carry objects even when they can still hold objects?
>
> We apologize for the confusion. A detailed analysis of Figure 6 can be found in Section 4.3, “Analysis For CooHOI Framework,” lines 283-302. Additional visualizations are available in Appendix D. To reiterate:
> - Without the “stand point” design, the agent sometimes fails to approach the shortest edge of the object and cannot hold it successfully.
> - Without the “dynamic observation” design, the agents would just stand in front of the object, seemingly unsure of what to do.
> - Without the “reverse walk” design, the agents may hold the objects but cannot reach the target position successfully, resulting in a deadlock.
> - Without “Initialization,” meaning if we train from scratch, the two-agent policy fails to achieve successful carrying. They will just stand near the objects, but don't know how to carry them to the destination.
>
> > Q4: In addition, r_carry is only defined in the supplementary material. I suggest including it in Section 3.2.2. It would be better to plot the standard deviations in Figure 6, given that you evaluate four seeds.
>
> Thank you for your detailed advice. We placed the formulation in the appendix due to the NeurIPS page limit, but we acknowledge that this caused some confusion. In the next version of our paper, we will include $r_{carry}$ in Section 3.2.2, following the definition of $r_{target}$. Additionally, we appreciate your suggestion to incorporate the standard deviation plot in Figure 6, and we will include it in the next version.
>
> > Q5: What is the action space of the policy? Is it the target joint position for each joint?
>
> Yes, the action space consists of the target joint position for each joint. This is then converted into force using PD control: `(target_pos - dof_pos) * stiffness - dof_vel * damping`. A detailed explanation of this can be found in our response to Reviewer TU4S Q1. Please refer to that if you have further questions. Note that this approach is a **common paradigm** adopted by the physics-based character animation community [1][2][3][4][5][6][7][8][9][10][11][12][13].

---

> ### Comment · Area_Chair_HGdc · 2024-08-12
> **To reviewer ehnN : Please respond to rebuttal**
>
> Hi reviewer ehnN ,
>
> Thank you for your initial review. Please kindly respond to the rebuttal posted by the authors.
> Does the rebuttal answer your questions/concerns? If not, why?
>
> Best,
> AC

---

### Author Rebuttal · Authors · 2024-08-06

We thank the reviewers for their detailed, valuable, and insightful feedback. We are pleased that the reviewers recognize our dedication to **addressing the under-studied and important application** of physics-based multi-agent cooperation HOI (Reviewer kufk). We appreciate the acknowledgment that our ideas on **utilizing manipulated object dynamics can provide valuable insights** for future research (Reviewer Zw8u). Our paper’s **comprehensive ablation experiments and boundary analysis** were noted positively (Reviewers ehnN, Zw8u, kufk, Tu4S), and the reviewers commended **our well-written paper** (Reviewers ehnN, Zw8u, kufk) and **well-designed framework** (Reviewers ehnN, kufk). Furthermore, our visualizations have been highlighted for demonstrating **natural HOI motion** (Reviewers ehnN, Tu4S).

We are pleased to report that we have conducted the analysis suggested by the reviewers on why the four agents failed to carry objects in our previous experiments. We have also designed a new scenario to bypass the limitation of the absence of dexterous hands, demonstrating the effectiveness and generalizability of our methods.

### Successful 4-Agent Coordination in Carrying an H-Shaped Box

In the limitations section, we discussed that our framework fails when four agents collaborate to carry big box. We attribute this failure to **the absence of dexterous hands and limited friction**. During the process of carrying a box to its destination, we observed that the large box sometimes slips from the humanoid agents’ hands. This is because the hand model of the humanoid agent is spherical, making it difficult to hold the corners of the box, which leads to the box being **“squeezed out” of their hands**.

However, we found a way to bypass this limitation without dexterous hands: we had the four agents carry an **“H-shaped” large box** and **successfully complete the task**. We report the success rate of our policy when carrying objects of different weights. When carrying 60kg H-shape box, our policy achieves 90.63% success rate and 23.56 cm precision. Considering the H-shaped object is relatively large (3m x 3m x 0.5m), we define success as having the center of the object within 0.5m of the target position. Please refer to **Fig1 and Fig2 in the attached PDF** for more visualizations. The success rate and precision of the 4-agent carrying policy are detailed below:


| Agent Number | Object Category | Weight of the Object (kg) | Success Rate (%) | Precision (cm) |
| ------------ | --------------- | ------------------------- | ---------------- | -------------- |
| 4            | H-shape Box     | 60                        | 90.63            | 23.56          |
| 4            | H-shape Box     | 70                        | 88.28            | 24.97          |
| 4            | H-shape Box     | 80                        | 76.95            | 24.72          |
| 4            | H-shape Box     | 90                        | 69.53            | 25.01             |


### References for All Reviewers

**Dear reviewers, due to the rebuttals' character limit, we've placed the references for all rebuttals below. Thank you for your time and consideration**.

[1] Hassan, Mohamed, et al. "Synthesizing physical character-scene interactions." *ACM SIGGRAPH 2023 Conference Proceedings*. 2023.

[2] Tessler, Chen, et al. "Calm: Conditional adversarial latent models for directable virtual characters." _ACM SIGGRAPH 2023 Conference Proceedings_. 2023.

[3] Xiao, Zeqi, et al. "Unified human-scene interaction via prompted chain-of-contacts." arXiv preprint arXiv:2309.07918 (2023).

[4] Peng, Xue Bin, et al. "Deepmimic: Example-guided deep reinforcement learning of physics-based character skills." ACM Transactions On Graphics (TOG) 37.4 (2018): 1-14.

[5] Peng, Xue Bin, et al. "Amp: Adversarial motion priors for stylized physics-based character control." ACM Transactions on Graphics (ToG) 40.4 (2021): 1-20.

[6] Juravsky, Jordan, et al. "Padl: Language-directed physics-based character control." _SIGGRAPH Asia 2022 Conference Papers_. 2022.

[7] Peng, Xue Bin, et al. "Ase: Large-scale reusable adversarial skill embeddings for physically simulated characters." ACM Transactions On Graphics (TOG) 41.4 (2022): 1-17

[8] Juravsky, Jordan, et al. "SuperPADL: Scaling Language-Directed Physics-Based Control with Progressive Supervised Distillation." _ACM SIGGRAPH 2024 Conference Papers_. 2024.

[9] Rempe, Davis, et al. "Trace and pace: Controllable pedestrian animation via guided trajectory diffusion." _Proceedings of the IEEE/CVF Conference on Computer Vision and Pattern Recognition_. 2023.

[10] Luo, Zhengyi, et al. "Perpetual humanoid control for real-time simulated avatars." _Proceedings of the IEEE/CVF International Conference on Computer Vision_. 2023.

[11] Wang, Yinhuai, et al. "Physhoi: Physics-based imitation of dynamic human-object interaction." arXiv preprint arXiv:2312.04393 (2023).

[12] Luo, Zhengyi, et al. "Grasping Diverse Objects with Simulated Humanoids." arXiv preprint arXiv:2407.11385 (2024).

[13] Luo, Zhengyi, et al. "SMPLOlympics: Sports Environments for Physically Simulated Humanoids." arXiv preprint arXiv:2407.00187 (2024).

[14] He, Tairan, et al. "Learning human-to-humanoid real-time whole-body teleoperation." arXiv preprint arXiv:2403.04436 (2024).

[15] He, Tairan, et al. "OmniH2O: Universal and Dexterous Human-to-Humanoid Whole-Body Teleoperation and Learning." arXiv preprint arXiv:2406.08858 (2024).

[16] Rudin, Nikita, et al. "Learning to walk in minutes using massively parallel deep reinforcement learning." _Conference on Robot Learning_. PMLR, 2022.

[17] Pan, Liang, et al. "Synthesizing physically plausible human motions in 3d scenes." _2024 International Conference on 3D Vision (3DV)_. IEEE, 2024.

[18] Zhang, Yunbo, et al. "Simulation and retargeting of complex multi-character interactions." _ACM SIGGRAPH 2023 Conference Proceedings_. 2023.

---

### Decision · Program_Chairs · 2024-09-25

**Decision:**

Accept (spotlight)

**Comment:**

This paper introduces CooHOI, a framework for learning cooperative human-object interactions in physics-based character animation. The method demonstrates the ability to learn multi-agent object transportation tasks using only single-agent motion capture data, with good generalization to objects of varying shapes and weights. All reviewers recommended acceptance, praising the paper's well-designed approach, comprehensive experiments, and valuable contribution to an under-studied area. The reviewers consistently highlighted the effectiveness of using manipulated object dynamics for implicit communication between agents and the framework's ability to transfer single-agent skills to multi-agent scenarios.

Initial concerns focused on the limited generalization to diverse object shapes, the use of simplified humanoid models without dexterous hands, and questions about the robustness of the system. The authors provided detailed responses, clarifying their approach to object diversity, explaining the rationale behind their design choices, and offering additional experiments demonstrating the framework's capabilities. These responses addressed many reviewer concerns, though some reservations remain about generalization and the limitations imposed by simplified hand models.

Despite these limitations, the consensus is that CooHOI represents a significant contribution to the field of physics-based character animation and multi-agent cooperation. The paper's thorough evaluations, clear presentation, and novel insights outweigh its current limitations. For the final version, I encourage the authors to incorporate their rebuttal explanations, particularly regarding object diversity and the rationale behind their design choices. Additionally, discussing potential future work on incorporating dexterous hands and improving generalization to more diverse objects would strengthen the paper further.